# Ferroptosis Inducer Improves the Efficacy of Oncolytic Virus-Mediated Cancer Immunotherapy

**DOI:** 10.3390/biomedicines10061425

**Published:** 2022-06-15

**Authors:** Weilin Liu, Hongqi Chen, Zhi Zhu, Zuqiang Liu, Congrong Ma, Yong J. Lee, David L. Bartlett, Zong-Sheng Guo

**Affiliations:** 1Department of Surgery, University of Pittsburgh School of Medicine, Pittsburgh, PA 15213, USA; 603302@csu.edu.cn (W.L.); hqchen08@163.com (H.C.); zhuzhi@cmu.edu.cn (Z.Z.); zuqiang.liu@ahn.org (Z.L.); mac@upmc.edu (C.M.); leeyj@upmc.edu (Y.J.L.); 2UPMC Hillman Cancer Center, Pittsburgh, PA 15232, USA; 3Xiangya Medical College, Central South University, Changsha 410013, China; 4AHN-Cancer Institute, Pittsburgh, PA 15212, USA; 5Department of Immunology, Roswell Park Cancer Institute, Buffalo, NY 14203, USA

**Keywords:** oncolytic virus, ferroptosis, stimulator, erastin, antitumor immunity, T cells, dendritic cells, combination, immunotherapy, efficacy

## Abstract

Ferroptosis is a type of programmed cell death dependent on iron and characterized by the accumulation of lipid peroxides. In this study, we explore the combination of a ferroptosis activator with an oncolytic vaccinia virus in tumor models. Erastin induced cell death in hepatoma, colon, and ovarian cancer cells, but not in melanoma cancer cells. Erastin, not the oncolytic vaccinia virus (OVV), induced the expression of key marker genes for ferroptosis in cancer cells. In hepatocellular carcinoma and colon cancer models, either erastin or OVV inhibited tumor growth, but a combination of the two yielded the best therapeutic effects, as indicated by inhibited tumor growth or regression and longer host survival. Immunological analyses indicate that erastin alone had little or no effect on systemic immunity or local immunity in the tumor. However, when combined with OV, erastin enhanced the number of activated dendritic cells and the activity of tumor-infiltrating T lymphocytes as indicated by an increase in IFN-γ^+^CD8^+^ and PD-1^+^CD8^+^ T cells. These results demonstrate that erastin can exert cytotoxicity on cancer cells via ferroptosis, but has little effect on immune activity by itself. However, when combined with an OVV, erastin promoted antitumoral immunity and efficacy by increasing the number of activated dendritic cells and promoting the activities of tumor specific CD8^+^ T cells in the tumor.

## 1. Introduction

A cancer immunotherapy breakthrough in the early 2010s, marked by objective clinical responses and longer survival in patients with advanced melanoma and other solid cancers, won the Nobel Prize in Medicine in 2018 [1]. However, only a minority of patients benefit from regimens using the dominant immune checkpoint blockade (ICB) approach. Many cancers are resistant to immunotherapy such as cancer vaccines or adoptive T cell therapy, as well as ICB [2,3,4]. 

Oncolytic virus (OV)-mediated cancer therapeutics are considered a novel type of immunotherapy [5,6,7]. OVs can selectively infect, replicate in, and kill cancer cells and/or tumor-associated stromal cells. One characteristic of OVs is that they often induce a form of cell death called immunogenic cell death (ICD) that eventually induces antitumor immunity [8]. As the tumor microenvironment (TME) is highly immunosuppressive, OV-elicited antitumor effects alone are often not sufficient to achieve therapeutic efficacy in advanced disease. To enhance the potential efficacy of oncolytic therapy, investigators designed various combination strategies to boost antitumoral immunity [5,9,10]. These strategies include, but are not limited to, expressing cytokines or co-stimulatory molecules to stimulate antitumor immunity [11,12,13] or combining strategies with other immunotherapy regimens such as ICB, cancer vaccine, or CAR T cells [9,14,15]. 

We developed highly tumor-selective and potent oncolytic vaccinia viruses (VVs) for cancer virotherapy [16]. An oncolytic VV typically works by inducing cancer cell death via necroptosis and apoptosis, thus inducing antitumoral immunity; this results in significant therapeutic efficacy in multiple tumor models in preclinical studies. One of our early recombinant oncolytic VVs is vvDD. This OV has two deleted viral genes that encode thymidine kinase and vaccinia growth factor for improved selectivity [16,17]. However, neither Pexa-Vec, the well-studied oncolytic VV, nor vvDD, achieved the desired clinical responses required for approval by the FDA or other agencies. The results from clinical trials reinforce the notion that OVs need to be improved or utilized in rational combinations for meaningful clinical efficacy in most solid tumors. There are two major issues that need to be overcome before more OVs can be approved to treat human patients. One is the limited efficacy of systemic administration of OV. In the case of VV, significant progress was made by multiple groups [18,19]. The second issue is therapeutic efficacy. Many approaches were developed to improve efficacy. OV armed with certain immunostimulatory genes is one such an approach [20]. The combination of OV with immune checkpoint blockade is the best studied combination approach [21,22]. Our current study represents such a combination. 

Over the last two decades, an interesting concept regarding the immunological consequence of cell death was established: ICD of cancer cells [23]. The ICD process is accompanied by the presentation of damage-associated molecular patterns (DAMPs), including the pre-apoptotic exposure of calreticulin (CRT) on the cell surface, the lysosomal secretion of ATP during apoptosis or other types of cell death, and the release of high mobility group box 1 (HMGB1) from dying/dead cells [23,24], leading to elicited antitumor immunity. A variety of regulated cell death modals, such as ferroptosis, necroptosis, and pyroptosis, are considered to be ICD and can elicit antitumor immune responses [25,26]. However, the case of ferroptosis is complex, as discussed in the next two paragraphs.

Ferroptosis is a type of cell death induced by the small molecules erastin or RSL3, as discovered by the Stockwell group [27]. It is characterized by a large amount of iron accumulation and lipid peroxidation during the cell death process; ferroptosis is iron-dependent [27,28,29]. Factors that induce ferroptosis can affect glutathione peroxidase through different pathways, which results in a reduction in antioxidant capacity and the accumulation of lipid reactive oxygen species in cells, ultimately leading to oxidative cell death [26,29]. Erastin is a small molecule capable of initiating ferroptotic cell death [27,30]. Erastin binds with and inhibits VDAC2 and VDAC3 and functionally inhibits the cystine-glutamate antiporter system Xc^-^. Cells treated with erastin are deprived of cysteine and are unable to synthesize the antioxidant glutathione. Ferroptosis is a regulated cell death nexus linking metabolism, redox biology, and disease, and may be useful as a target for therapeutics [31].

Ferroptosis’ functions and mechanisms of action in antitumor immunity remain to be elucidated. These functions and mechanisms may come from two groups of target cells: immune cells and cancer cells, respectively. On immune cells, a recent study by Wang et al. provided solid evidence that CD8^+^ T cells regulate tumor ferroptosis during cancer immunotherapy [32]. However, CD36-mediated ferroptosis dampens intratumoral CD8^+^ T cell effector function and impairs its antitumor ability [33]. In addition, PPARG-mediated ferroptosis in dendritic cells limits antitumor immunity [34]. On tumor cells, as ferroptosis is a form of ICD and can release various DAMPs, it is thought to be immunogenic when cancer cells are induced to undergo ferroptosis. Indeed, vaccination with early ferroptotic cancer cells induces efficient antitumor immunity [35]. These studies provide a complex picture of the roles of ferroptosis in antitumor immunity [26,36]. The evidence so far suggests that ferroptosis in cancer cells is good, but ferroptosis in immune cells is bad for antitumor immunity. 

What a ferroptosis activator might do to a tumor and intact immune system, and if it can work synergistically or additively with an OV for improved immunotherapy, remain largely open questions. In this study, we hypothesize that an oncolytic VV, when combined with the ferroptosis activator erastin, may have additive, if not synergistic, effects for cancer therapy. A superagonist IL-15-armed oncolytic VV was used in this study simply because it is one of the most potent OVs in inducing antitumor immunity and therapeutic results [12]; we aimed to achieve the best possible therapeutic results. Our results show that erastin, but not the OV, induced cancer cell ferroptosis in certain cancer cells in vitro. In tumor models, both monotherapies displayed some therapeutic effects, but the combination yielded the best therapeutic efficacy and long-term immune memory. Immunological analyses demonstrate that erastin alone induced little change in systemic and local immunity for cancer, but combined with the OV, it increased the number of activated dendritic cells and enhanced the activity of tumor specific CD8^+^ T cells. These targeted immunological effects may explain the improved therapeutic efficacy.

## 2. Materials and Methods

### 2.1. Cell Lines

B16, Hepa1-6, MC38-luc, and ID8-luc tumor cells were all used in our previous studies. B16 (CRL-6322) and Hepa1-6 (CRL-1830) were originally obtained from American Type Culture Collection (Manassas, VA, USA). MC38 is a murine colon adenocarcinoma cell line from a C57BL/6 mouse [37]. We modified this cell line by transducing with a lentivirus expressing firefly luciferase gene, resulting in MC38-Luc [38]. The ID8 cell line was a gift from Dr. Katherine F. Roby, who with her coworkers developed this murine ovarian cancer cell line [39]. We also tagged it with firefly luciferase (ID8-luc). All these lines of cells were grown in Dulbecco’s Modified Eagle’s Medium supplemented with 100 U penicillin and 100 µg/L streptomycin (Invitrogen, Carlsbad, CA, USA), 2 mM L-glutamine, and 10% fetal bovine serum (FBS) (R&D Systems, Minneapolis, MN, USA) in an incubator at 37 °C with 5% CO_2_.

### 2.2. Chemicals and Small Molecule Inhibitor Erastin

Erastin was obtained from ApexBio (Houston, TX, USA). The chemical was stored at −20 °C and dissolved in DMSO as stock solution. The stock solution was diluted with 1× phosphate buffered saline (PBS) to appropriate concentration before use.

### 2.3. Oncolytic Viruses vvDD and vvDD-IL15Rα

These OVs were described and used in previous studies [12,40]. The preparation, storage, and tittering of the viruses were described in our previous study [12].

### 2.4. Cell Viability Assays

Hepa1-6 (1.0 × 10^4^ cells), MC38-luc (1.0 × 10^4^ cells), ID8 (5.0 × 10^3^ cells), and B16 (5.0 × 10^3^ cells) were plated per well in 96-well plates. The next day, cells were cocultured with different concentrations of erastin (0, 0.25, 1.0, 2.5, 5.0 µM). Cell viability was measured after treatments for 24 h using MTS assay (Promega, Madison, MI, USA).

### 2.5. Mice and Murine Tumor Models

C57BL/6J (B6) female mice were purchased from the Jackson Laboratory (Bar Harbor, ME, USA) and housed in the University of Pittsburgh Animal Facility under pathogen-free conditions. The Institutional Animal Care and Use Committee at the University of Pittsburgh approved all animal studies and procedures.

B6 mice were injected with 2.0 × 10^6^ Hepa1-6 tumor cells or 5.0 × 10^5^ MC38-Luc tumor cells in 60 µL volume subcutaneously (s.c.) on the right flank to establish tumor implants and were randomly divided into required groups. When tumor nodules reached ~5 × 5 mm, vvDD-IL15-Rα were intratumorally (i.t.) injected with 2.0 × 10^7^ pfu/50 µL for only one injection and 20 mg/kg erastin was intraperitoneally (i.p.) injected daily for seven total injections. At some timepoints, mice were sacrificed to harvest s.c. tumor nodules as well as spleens for analyses. As for the rechallenge assay, naïve mice and B6 mice cured by dual therapy were s.c. injected with 2.0 × 10^6^ Hepa1-6 tumor cells; the tumor formation rate was recorded after 12 days. Tumor growth was monitored via digital caliper every other day and tumor volume was calculated as: V (mm^3^) = 0.5 × (Length × Width^2^).

Animal health status and survival was monitored closely. Mice were sacrificed when tumors reached 2000 mm^3^ in size, became ulcerated, and/or interfered with murine activity.

### 2.6. Analysis of Immune Cells in the Tumor Microenvironment

B6 mice were injected with 5.0 × 10^5^ MC38-Luc tumor cells in 60 µL volume s.c. on the right flank to establish tumor implants and randomly divided into required groups. Data are quantified (*n* = 5 in the PBS and vvDD-IL15-Rα groups; *n* = 6 in the erastin and dual therapy groups). When tumor nodules reached ~5 × 5 mm, vvDD-IL15-Rα were i.t. injected with 2.0 × 10^7^ pfu/50 µL for only one injection and 20 mg/kg erastin was i.p. injected daily for seven total injections. Tumor tissues were recovered eight days after the first treatment for further analysis.

### 2.7. Flow Cytometry

We collected spleens and tumor tissues, weighed, and incubated them in RPMI 1640 medium that contained 2% FBS, 1 mg/mL collagenase IV (Sigma (St. Louis, MO, USA) #C5138), 0.1 mg/mL hyaluronidase (Sigma #H6254), and 200 U/mL DNase I (Sigma #D5025) at 37 °C for 1–2 h to make single cell preparations. Following this, these single cells were blocked with α-CD16/32 Ab (eBioscience, Thermo Fisher Scientific, Waltham, MA, USA) (14-0161-85; clone 93; dilution 1:1000), then stained with antibodies against mouse CD45 (PerCP-Cy5.5; clone 30-F11; BioLegend (San Diego, CA, USA) #103132; dilution 1:300), CD3 (FITC; clone 17A2; eBioscience #11-0032-82; dilution 1:300), CD4 (FITC; clone GK1.5; BD Biosciences (Franklin Lakes, NJ, USA) #553729; dilution 1:300), CD8 (FITC; clone 53-6.7; BioLegend #100706; dilution 1:300), PD-1 (PE; clone J43; eBioscience #12-9985-83; dilution 1:300), CD206 (PE; clone 19.2; eBioscience #12-2069-42; dilution 1:20), CD86 (FITC; eBioscience #11-0862-82; dilution 1:300); CD11b (FITC; clone M1/70; BioLegend #101206; dilution 1:300), CD11c (APC; clone N418; eBioscience #17-0114-82; dilution 1:300), CTLA-4 (APC; clone UC10-4B9; eBioscience #17-1522-82; dilution 1:300), F4/80 (APC; clone BM8; Fisher Scientific #50-112-9524; dilution 1:300), Foxp3 (PE; clone FJK-16s; eBioscience #12-5773-82; dilution 1:100), Gr-1 (APC; clone RB6-8C5; eBioscience #17-5931-82; dilution 1:300), IFN-γ (APC; clone #1819 XMG1.2; eBioscience #17-7311-82; dilution 1:100), and NK1.1 (APC; clone PK136; BioLegend #108710; dilution 1:300). The intracellular FOXP3 and IFN-γ staining kits were obtained from BioLegend. A BD Accuri C6 cytometer was used for flow cytometry, and the machine associated software was used to analyze and process the data.

### 2.8. RT-qPCR

RT-PCR was performed according to the previously described procedure [41]. Briefly, total RNA was isolated from tumor tissue or cell lysate using RNeasy kit (Qiagen, Germantown, MD, USA). cDNA was then synthesized in a reaction with 2.0 μg of RNA using qScript^™^ cDNA SuperMix (Quanta Biosciences, Inc., Gaithersburg, MD, USA) and a Dyad^®^ Peltier Thermal Cycler (Bio-Rad, Hercules, CA, USA). qPCR was then performed using TaqMan analysis with PerfeCTa^®^ qPCR SuperMix (Quanta Biosciences, Inc.) and a StepOnePlus System (Life Technologies, Grand Island, NY, USA). All primers for PCR were ordered from a commercial source (Thermo Fisher Scientific; Waltham, MA, USA). Relative gene expression was calculated by comparing to a housekeeping gene, hypoxanthine-guanine phosphoribosyltransferase (HPRT1) as fold increase (2^−ΔCT^), where ΔCT = CT _(Target gene)_ − CT _(HPRT1)_.

### 2.9. IFN-γ ELISpot Assays

Spleens were collected from MC38-luc tumor-bearing mice at 8 days post virus administration. These isolated splenocytes (at 1.0 × 10^6^ cells/mL) were re-stimulated with 200-Gy irradiated MC38-luc or ID8 cells (2.0 × 10^5^ cells/mL) in 200 μL RPMI 1640 medium supplemented with 10% FBS. After incubation at 37 °C, 5% CO_2_ for 24 h, microplates were thoroughly washed before incubating with biotinylated α-mouse IFN-γ Ab overnight (mAb R4-GA2-Biotin; Mabtech, Inc., Cincinnati, OH, USA). These plates were then processed using the protocol provided by the vendor (cat no. SK-4200; Vector Laboratories, Inc., Burlingame, CA, USA). Data were collected and analyzed using an ImmunoSpot™ analyzer (Cellular Technology, Ltd., Shaker Heights, OH, USA).

### 2.10. Statistical Analysis

Raw data were processed using GraphPad Prism software (GraphPad Software, San Diego, CA, USA). One-way and two-way ANOVA were used to analyze the collected data and presented as means ± SD whenever suitable. Kaplan-Meier survival analysis and the log-rank test were used to analyze anima survival data. A *p* value of <0.05 was considered statistically significant. Standard symbols for statistical significance described in figures were used as follows: * *p* < 0.05; ** *p* < 0.01; *** *p* < 0.001; **** *p* < 0.0001; and ns: not significant.

## 3. Results

### 3.1. Erastin Exhibits Various Extents of Cytotoxicity to Different Cancer Cells

We first explored the cytotoxicity of erastin to some murine cancer cells (Figure 1) representing hepatoma (Hepa1-6), colon cancer (MC38-luc), ovarian cancer (ID8), and melanoma (B16). We tested the concentrations of erastin from 0, 0.25, to 5.0 µM. We found that Hepa1-6, MC38-luc, and ID8 cancer cells were all sensitive to erastin-induced ferroptosis and cell death. The values of ID50 were approximately 1.2 µM for Hepa1-6 cells, 0.5 µM for MC38-luc cells, and 1.2 µM for ID8 cells. In contrast, B16 cancer cells were highly resistant to the induction of ferroptosis, at least up to 5.0 µM, the maximal dose tested in this experiment.

### 3.2. The Analysis of Expression of Marker Genes for Ferroptosis

There are currently no direct assays to distinguish between different forms of non-apoptotic cell death. However, the distinct regulatory pathways involved with each provide distinct protein markers that can be used for their detection. Based on previous studies, we chose four genes as ferroptosis markers in this study: heme oxygenase 1 (Hmox1), ferritin light polypeptide 1 (FTL1), ferritin heavy chain 1 (FTH1), and NAD(P)H quinone dehydrogenase 1 (Nqo1) [42,43,44]. We examined the expression of these four cellular genes in cancer cells that were mock-treated or treated with the OV, erastin, or the dual treatment (Figure 2). MC38-luc cancer cells were treated for 24 h, cells were harvested, and total RNAs were made from the cells; then RT-qPCR were performed for expression of specific genes at the mRNA level. Erastin treatment induced the mRNA expression levels of Hmox1, FTL1, and FTH1 (**, ***, and * compared to controls, respectively). In addition, there was a tendency for Nqo1 upregulation (*p* = ns). The combination of erastin and OV treatment maintained the induction of these three marker genes (Figure 3). Interestingly, when we examined the marker gene expression in the OV-treated cells, we also observed the upregulation of Hmox1 and FTH, but not FTL1 and Nqo genes (Appendix A). Together, these results may suggest that erastin indeed induced ferroptosis while the OV did not.

### 3.3. The Therapeutic Effects of Erastin, OV, or Their Combination on Hepatoma and Colon Cancer Models in Syngeneic Mice

We then tested the hypothesis that the combination of erastin and OV-IL15-Rα will enhance therapeutic efficacy in the Hepa1-6 tumor model. As a first step, we needed to test the OV and the combination in this cancer cell line in vitro (Figure 4A). The OV induced dose-dependent cytotoxicity on Hepa1-6 cancer cells. When combined with erastin, the dual therapy resulted in more cytotoxicity in an erastin dose-dependent manner. Next, we examined its efficacy and safety in a Hepa1-6 s.c. tumor model (Figure 4B,C). Tumor models were established and treated according to a schedule as described (Appendix A). The tumor volumes were plotted by individual mouse (Figure 4B). Erastin alone did inhibit tumor growth (four out of five mice had a significant reduction in tumor volume). In the group treated with vvDD-IL-15Rα, two out of five mice had a reduction in tumor volume while the other three mice had complete tumor regression. However, the dual therapy led to complete tumor regression in five out of five mice (*p* ≤ 0.05 compared to vvDD-IL-15-Rα; *p* ≤ 0.0001 when compared to other groups). The capacity of tumor growth inhibition translated as longer mouse survival in the groups (Figure 4C). Not surprisingly, 100% of mice treated with the dual therapy survived for up 100 days, the total duration of the experiment (* compared to vvDD-IL15-Rα; ** compared to erastin alone). To see if an immune memory was present in these cured mice, we performed a rechallenge experiment with Hepa1-6 cancer cells (Figure 4D). When observed within 12 days, zero out of five cured mice grew a tumor, while six out of seven naïve mice grew a tumor. These results indicated immune memory in the Hepa1-6 tumor-bearing mice cured by the combination treatment.

We conducted the same type of therapeutic experiment in another erastin-sensitive MC38-luc colon cancer model (Figure 5). Again, each single agent treatment led to tumor inhibition (*p* < 0.001, compared to PBS), yet the dual therapy obtained the best therapeutic efficacy assessed as inhibition of tumor growth (*p* ≤ 0.05 when compared to OV alone; *p* < 0.001 when compared to erastin group) (Figure 5A). These inhibitory effects of tumor growth also translated into longer mouse survival, with the best result obtained from the dual therapy (Figure 5B).

### 3.4. The Dual Therapy Induced More Potent Antitumor Immunity

In order to examine if the antitumor effects occur through immunological mechanisms, we set up another experiment using MC38-luc tumors. Eight days after therapy, tumors were harvested and weighed (Figure 5C). The tumor weights in the groups confirmed that each monotherapy was effective, and that the dual therapy was the most effective. Following this, spleens were collected, splenocytes were isolated, and IFN-γ ELISpot assay was performed (Figure 5D). When these splenocytes were re-stimulated with irradiated ID8 cells (control), few spots were observed in all groups. When the splenocytes were re-stimulated with parental (irradiated) MC38-luc cancer cells, a different pattern was observed. In the groups treated with PBS or erastin, few spots were observed. However, in the vvDD-IL15-Rα group, approximately 5 spots/1.0 × 10^5^ splenocytes (or ~50 spots/1.0 × 10^6^ splenocytes) were observed. In the dual treatment group, this number went up to 120/1.0 × 10^6^ splenocytes (*p* < 0.05 compared to vvDD-IL-15-Rα; *p* < 0.001 when compared to the PBS group). Altogether, these data show that the dual treatment induced a significant tumor-specific antitumor memory immune response.

Using RT-qPCR assays (Figure 6) and later flow cytometry (Figure 7), we conducted detailed analyses of immunological profiles in the tumor tissues. Using RT-PCR (Figure 6), the first observation was that erastin had little effect on any of the immune cells, as the values were all similar to those in the PBS group. The OV stimulated most types of adaptive immune cells as expected, including both CD4 and CD8 cells, as well as mRNAs for Th1 cytokines IFN-γ and TNF-α. In the dual treatment group, these markers were as high as in the OV group, or in the case of IFN-γ, even higher than in the OV group (Figure 6C). However, both PD-1 and CTLA-4 were also elevated in the groups treated only with OV or the dual treatment. The immune checkpoint molecules PD-1 and CTLA-4 are markers for immune activation and then immune tolerance. 

When the RT-qPCR data panel was analyzed, erastin had little effect on immune cells, but vvDD-IL15-Rα had a large of impact on immune cells, as expected. The OV treatment increased CD4^+^ cells, CD8^+^ cells, expression of IFN-γ and TNF-α, and increased PD-1 and CTLA-4. When treatment only with OV was compared to the dual therapy, the only major difference was that IFN-γ and PD-1 in CD8^+^ T cells were further increased in dual therapy (*p* < 0.05).

These patterns were confirmed and extended using flow cytometry analyses (Figure 7). We examined the subsets of IFN-γ^+^CD8^+^ T cells, PD-1^+^CD8^+^ T cells, CD86^+^CD11c^+^ cells, macrophages, MDSC, and the ratio of CD8^+^ T cells/FoxP3^+^CD4^+^ T cells. We did not observe any changes induced by erastin alone, yet all positive changes involved vvDD-IL15-Rα. When the two were combined, we observed further enhancement in IFN-γ^+^CD8^+^ T cells (Figure 7A), PD-1^+^CD8^+^ T cells (Figure 7B), CD86^+^CD11c^+^ cells (Figure 7C), macrophages (Figure 7D), and MDSCs (Figure 7E), but not in the ratio of CD8^+^ T cells/FoxP3^+^CD4^+^ T cells (Figure 7F). In summary, the OV stimulated most types of “good” immune cells as expected. The OV increased both CD86^+^CD11c^+^ dendritic cells and macrophages, and dual therapy increased both of them further. Both OV therapy and dual therapy improved the ratio of CD8^+^ T cells/Foxp3^+^CD4^+^ T cells (Appendix A).

## 4. Discussion

The ferroptosis mode of cell death is a recent discovery [27], and its biological effects remain to be studied and explored. As for the consequence of eliciting immunity, a number of cell death modes can be classified as ICD [23,24]. However, whether ferroptosis enhancers can enhance immunogenicity of cancer and antitumor immunity in a therapeutic setting has not yet been investigated. It is a significant study to undertake [45].

In this study, we took a step forward in addressing these questions and explored the potential of combining a ferroptosis activator with a Th1-cytokine-armed OV for cancer immunotherapy in tumor models in syngeneic mice. Our results indicate that a number of cancer cell lines are susceptible to the ferroptosis activator, erastin, including cancer cell lines of HCC (Hepa1-6), colon cancer (MC38), and ovarian cancer (ID8). Not surprisingly, we found that one of the cell lines tested, melanoma B16, was resistant to the induction of ferroptosis by erastin. Can VV induce ferroptosis? Previous studies indicate that oncolytic VV induced cell death through dual modes ofapoptosis and necroptosis [46,47], the latter of which is ICD. Here, we showed for the first time that an oncolytic VV did not induce ferroptosis in cancer cells. However, another type of OV, Newcastle-disease virus, can induce ferroptosis in infected tumor cells [48]. It is interesting to note that ferroptosis and apoptosis can interact through certain signaling pathways, and inducers for apoptosis and ferroptosis may lead to improved tumoricidal efficacy [49].

In tumor models of syngeneic mice, we showed that either erastin or the OV exerted therapeutic efficacy on HCC and colon cancer. However, much improved efficacies were obtained when the two anti-tumor agents were combined on colon cancer and HCC. The Hepa1-6 tumor-bearing mice cured by vvDD-IL15-Rα and erastin were resistant to the rechallenge of Hepa1-6 cancer cells, suggesting that immune memory existed in these cured mice.

We then conducted immunological analyses. First, tumor-specific T cells in splenocytes were identified using ELISpot assays. We detected few if any tumor-reactive T cells in the spleens of mice treated with erastin. However, we detected a significant number of tumor-reactive T cells in the group treated with vvDD-IL15-Rα, and even more in the group that received the combination treatment. When flow cytometry and RT-qPCR analyses were conducted for immune cells isolated from the tumor tissues, we found that erastin by itself did not change the number of activated immune cells, including macrophages, dendritic cells, myeloid-derived suppressor cells (MDSCs), and CD4^+^ and CD8^+^ T cells. However, the OV impacted the immune profile of the tumor tissues, mostly for pro-antitumor immunity. Interestingly, the addition of erastin to OV, i.e., the combination, made two types of significant changes. One significant change was an increase in macrophages and activated dendritic cells. The second was that even though no changes in the numbers of CD4^+^ and CD8^+^ T cells were observed, these CD8^+^ T cells were more active, as indicated by increasing IFN-γ and PD-1 expression. In addition, the ratio of CD8^+^ T cells over Treg increased, which is a good indicator of stronger antitumor immunity. In summary, both innate and adaptive immunity against cancer were promoted by the combinatorial treatment of erastin and OV.

However, as quantities of CD4^+^ and CD8^+^ T cells and production of Th1 cytokines IFN-γ and TNF-α in the TME are increased after therapy by the OV or dual therapy with OV and erastin, quantities of MDSCs are also increased, as well as PD-1 and CTLA-4, the checkpoint molecules that can inhibit antitumor immunity. Our data show that PD-1^+^CD8^+^ T cells were increased by the OV treatment, and further promoted by the dual therapy (Figure 7). We did not analyze the CTLA-4-positive cell subpopulations in the current study. However, previous studies with OV treatment by us and others show that CTLA-4 and PD-1/PD-L1 can be expressed on CD4^+^ and CD8^+^ T cells, and other immune cells such as NK and macrophages [50,51,52,53,54]. In a future study using this dual combination regimen, we need to investigate the profile of checkpoint molecules’ expression in various subsets of immune cells and produce rational combinations of several ICIs with current dual therapy for improved therapeutic efficacy.

There are some studies on ferroptosis in cancer cells and signaling pathways, and what types of drugs can induce ferroptosis [26]. So far, few investigations have been undertaken to show the direct crosstalk between ferroptosis and antitumoral immunity. A biologically plausible hypothesis is that dying cells communicate with immune cells through a set of signals, such as the “find me” and “eat me” signals produced during the process of cell death. This is well studied in apoptosis, but not in other modes of cell death [55]. Few, if any, studies address whether tumor cells undergoing ferroptosis can enhance antitumor immunity in vivo. As for the function of ferroptosis in immune cells, two studies show that ferroptosis can dampen DC and CD8^+^ T cell functions [33,34]. Interestingly, Wang et al. reported that CD8^+^ T cells induced ferroptosis in cancer cells in vivo [32]. The CD8^+^ T cells, activated through immunotherapy, downregulated the expression of SLC7A11 protein, which is required for the induction of ferroptosis. Further in vivo experiments revealed that T cells induce ferroptosis in cancer cells in ovarian tumor-bearing mice [32].

Our current study, to our knowledge, is the first one to directly apply a ferroptosis enhancer in vivo in combination with an OV and observe subsequent induction of antitumor immunity and improved therapeutic efficacy. We made two new key discoveries. The first is that induction of ferroptosis in cancer cells by itself changed little, if any, systemic or local immunity in the TME. This seemed to be contradictory to the previous observations made by others [33,34]. However, our observation is from the net immunological effect in the TME. Its effects on immune cell subpopulations require further study. Second, when combined with OV, which elicited antitumor immunity, a ferroptosis enhancer promoted antitumor immunity through two major immunological effects. One was promoting activated dendritic cells in the tumor tissue, and the other (which may be an indirect effect of the first) was enhancing the activity of antitumor CD8^+^ T cells without increasing the number of these T cells in the TME. Our new findings suggest that this novel combination strategy is a promising therapeutic regimen that is warranted for clinical studies in human cancer patients.

## 5. Conclusions

Both erastin and OV show limited effects in eliciting antitumor immunity and therapeutic efficacy at the indicated doses in susceptible cancer models when used on their own. When combined, they work together to enhance immunotherapeutic efficacy by increasing tumor-specific IFN-γ^+^CD8^+^ and PD-1^+^CD8^+^ T cells. This combination regimen may become a novel strategy for cancer therapy. This combination also results in increased expression of PD-1 and CTLA-4 in the TME, providing the molecular basis for rational multiple combinations with ICB therapy in the future.

## Figures and Tables

**Figure 1 biomedicines-10-01425-f001:**
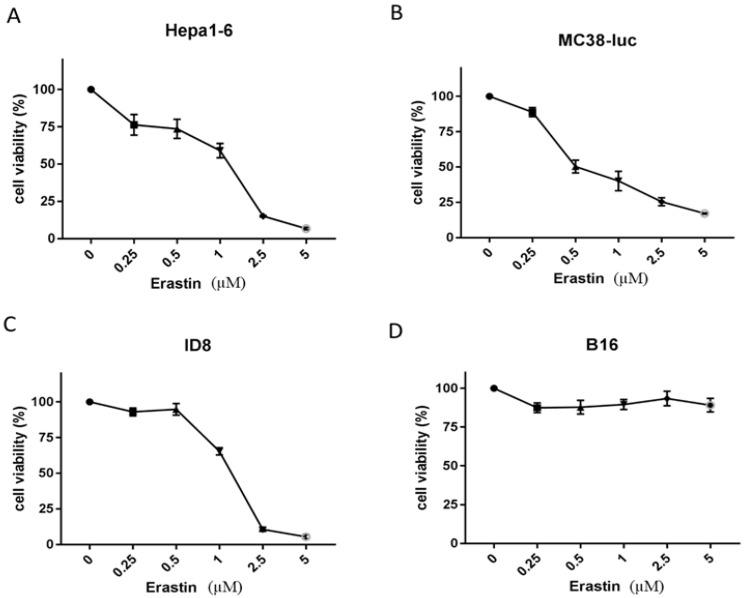
Erastin exhibits different cell toxicities on different cancer cell lines. Hepa1-6 (1.0 × 10^4^ cells) (**A**), MC38-luc (1.0 × 10^4^ cells) (**B**), ID8 (5.0 × 10^3^ cells) (**C**), and B16 (5.0 × 10^3^ cells) (**D**) were plated per well in 96-well plates and cocultured with different concentrations of erastin the next day as indicated. Cell viability was measured after treatment for 24 h using MTS assay. Data are presented as mean ± SD.

**Figure 2 biomedicines-10-01425-f002:**
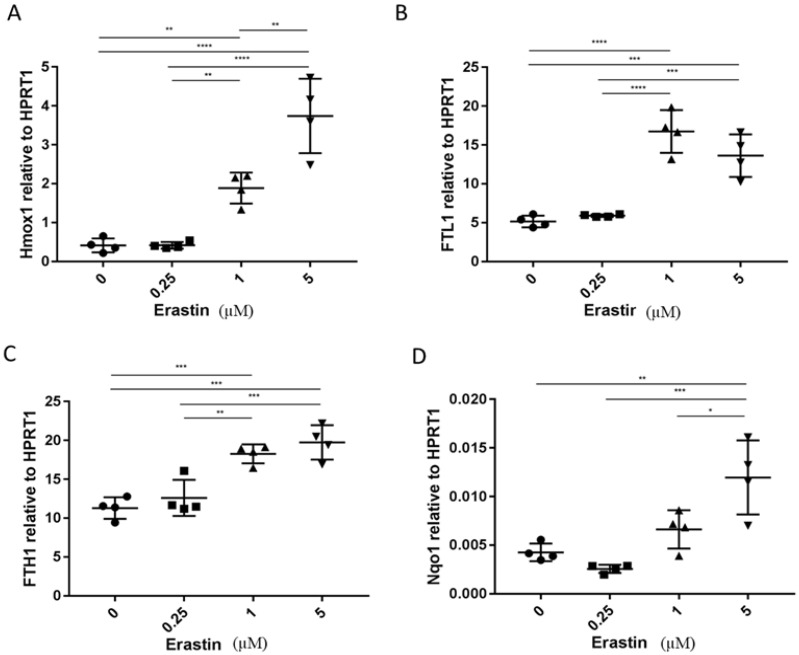
Erastin induced ferroptosis markers in a dose-dependent manner. MC38-luc (6.0 × 10^4^ cells) were plated in a 24-well plate and cocultured with different concentrations of erastin (0, 0.25, 1.0, 2.5 µM). After 24 h, cells were collected and cell lysates were used to extract RNA to determine the expression levels of Hmox1, FTL1, FTH1, and Nqo1 (**A**–**D**). The expression levels of all markers are relative to the housekeeping gene HPRT1. Values are mean ± SD. One-way ANOVA was used to analyze the statistical significance (*, *p* < 0.05; **, *p* < 0.01; ***, *p* < 0.001; ****, *p* < 0.0001).

**Figure 3 biomedicines-10-01425-f003:**
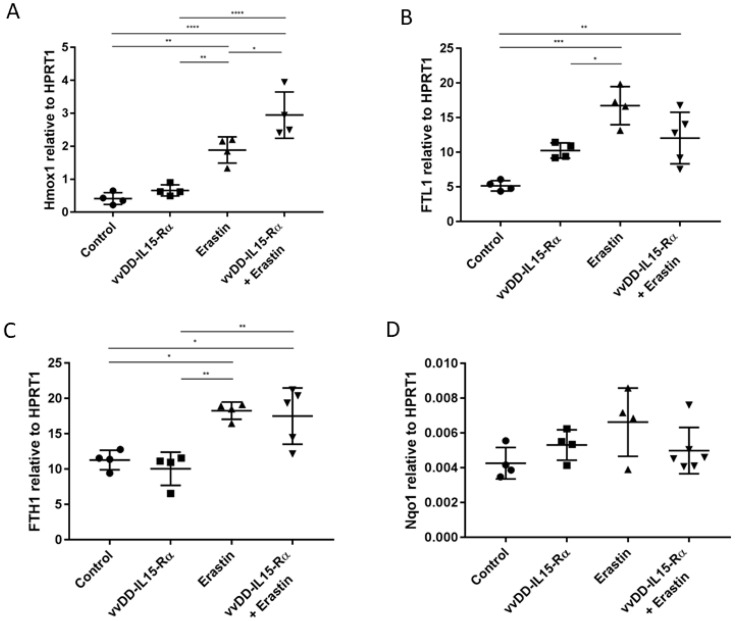
The combination of erastin and vvDD-IL-15-Rα also induced ferroptosis markers in MC38-luc colon cancer cells. MC38-luc cells were plated at 6.0 × 10^4^/well in a 24-well plate and infected with vvDD-IL15-Rα (MOI = 0.1) and/or erastin (1.0 µm) the next day. After 24 h, cells were collected and cell lysates were used to extract RNA to determine the expression levels of Hmox1, FTL1, FTH1, and Nqo1 (**A**–**D**). The expression levels of all markers are relative to the housekeeping gene HPRT1. Values are mean ± SD. One-way ANOVA was used to analyze the statistical significance (*, *p* < 0.05; **, *p* < 0.01; ***, *p* < 0.001; ****, *p* < 0.0001).

**Figure 4 biomedicines-10-01425-f004:**
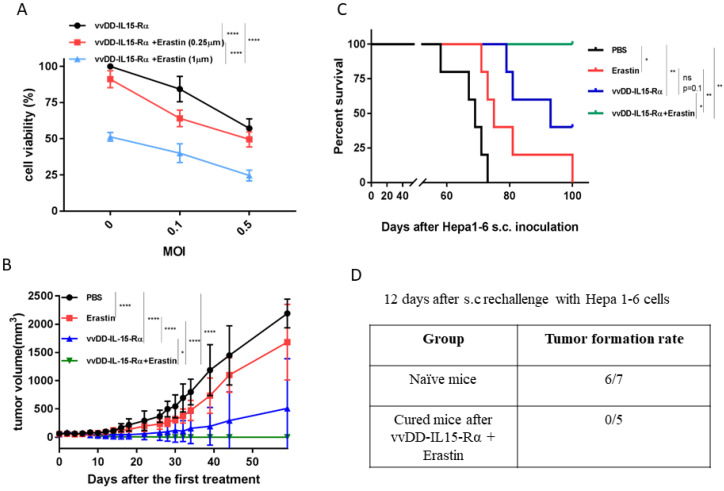
Combination therapy has superior anti-tumor response on Hepa1-6 tumor model. (**A**) Hepa1-6 cancer cells were plated at 1.0 × 10^4^/well in a 96-well plate and infected with vvDD-IL15-Rα (MOI = 0, 0.1, 0.5) and/or erastin (0.25, 1.0 µm) the next day. Cell viability was determined at 24 h after treatment using MTS assay. (**B**–**D**) B6 mice were inoculated subcutaneously (s.c.) with 2.0 × 10^6^ Hepa1-6 tumor cells in 60 µL volume and randomly divided into required groups (*n* = 5 in each group). When tumor nodules reached ~5 × 5 mm, vvDD-IL15-Rα were intratumorally (i.t.) injected with 2.0 × 10^7^ pfu/50 µL for only one injection; erastin was intraperitoneally (i.p.) injected daily with 20 mg/kg for seven total injections. (**B**) The graphs represent a comparison of tumor growth in different treatment groups. (**C**) The survival of tumor-bearing mice was monitored using Kaplan–Meier analysis and statistical analyses were performed using a log rank test. (**D**) Tumor formation rate on day 12 post rechallenge s.c. with 2.0 × 10^6^ Hepa1-6 cells. Two-way ANOVA was used to analyze the data (*, *p* < 0.05; **, *p* < 0.01; ****, *p* < 0.0001; and ns: not significant). This experiment was performed once.

**Figure 5 biomedicines-10-01425-f005:**
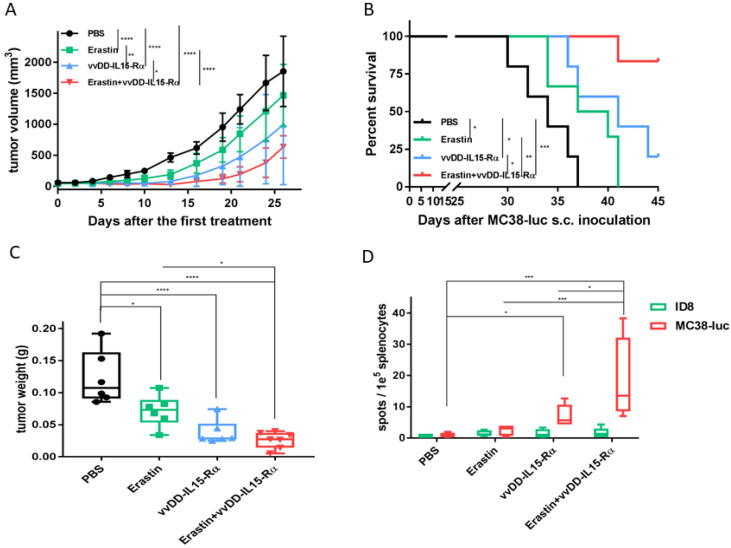
Combination therapy improved therapeutic efficacy and elicited potent anti-tumor immunity in MC38-luc tumor model. (**A**,**B**) B6 mice were inoculated subcutaneously with 5.0 × 10^5^ MC38-luc tumor cells in 60 µL volume and randomly divided into required groups. Data are quantified (*n* = 5 in PBS and vvDD-IL15-Rα groups; *n* = 6 in erastin and dual therapy groups). When tumor nodules reached ~5 × 5 mm, vvDD-IL15-Rα were intratumorally (i.t.) injected with 2.0 × 10^7^ pfu/50 µL for only one injection, erastin was intraperitoneally (i.p.) injected daily with 20 mg/kg for seven injections. (**A**) The graph represents a comparison of tumor growth in different treatment groups. (**B**) The survival of tumor-bearing mice was monitored using Kaplan–Meier analysis and statistical analyses were performed using a log rank test. (**C**,**D**) Another group of B6 mice was treated with the same time schedule as A and B; tumor nodules and spleens were collected and analyzed eight days post treatment. (**C**) Tumor nodules were collected and weighed. (**D**) ELISpot assay resulted in quantities of IFN-γ spots in the different treatment groups. Values are presented as mean ± SD. Two-way ANOVA was used to analyze the data in (**A**,**B**,**D**). One-way ANOVA was used to analyze data from panel **C** (*, *p* < 0.05; **, *p* < 0.01; ***, *p* < 0.001; ****, *p* < 0.0001). Parts of this experiment were performed three times and similar results were obtained.

**Figure 6 biomedicines-10-01425-f006:**
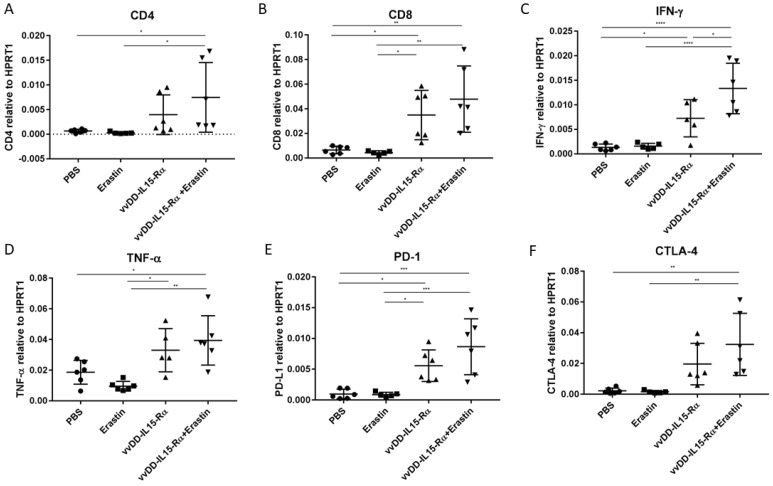
Tumor microenvironment (TME) is dynamically modified post combination therapy. Tumors are from the experiment depicted in Figure 5C, which were used to isolate RNA for RT-qPCR analyses. Expression levels of all markers are expressed relative to the housekeeping gene HPRT1. The expression levels of (**A**) CD4, (**B**) CD8, (**C**) IFN-γ, (**D**) TNF-α, (**E**) PD-1, and (**F**) CTLA-4 are presented. The symbols for the statistics are, *, *p* < 0.05; **, *p* < 0.01; ***, *p* < 0.001; ****, *p* < 0.0001.

**Figure 7 biomedicines-10-01425-f007:**
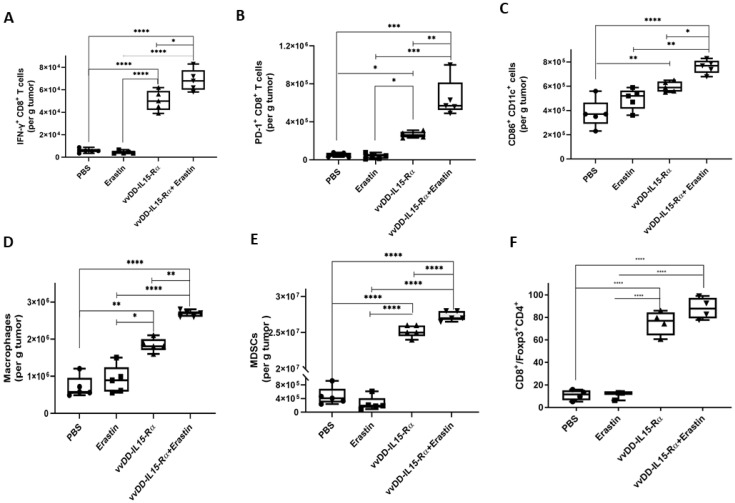
Tumor microenvironment (TME) is dynamically modified post combination therapy as analyzed by flow cytometry. Tumors are from the experiment depicted in Figure 5C, which were digested for FACS. (**A**) The quantities of activated IFN-γ^+^CD8^+^ T cells; (**B**) PD-1^+^CD8^+^ T cells; (**C**) activated CD86^+^CD11c^+^ cells; (**D**) macrophages (defined as CD45^+^CD11b^+^ F4/80^+^ CD206^+^); (**E**) myeloid-derived suppressor cells (defined as CD45^+^CD11b^+^Gr-1^+^); and (**F**) The ratio of CD8^+^/Treg (defined as CD45^+^CD4^+^FOXP3^+^) cells. *, *p* < 0.05; **, *p* < 0.01; ***, *p* < 0.001; ****, *p* < 0.0001.

## Data Availability

All supporting data are presented in the manuscript including the Appendix A.

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
