# Peer review of "Ferroptosis Inducer Improves the Efficacy of Oncolytic Virus-Mediated Cancer Immunotherapy"

_biomedicines, 2022, doi:10.3390/biomedicines10061425_

Round 1

Reviewer 1 Report

The study by Liu et al aims to assess the combined effect of oncolytic virus and ferroptosis-induced death on anti-tumour immunity. Whilst the topic is interesting, the manuscript is not clearly defined and presents major flaws that must be addressed:

Data and rationale

Overall, the figures are easy to read, however the results could be improved by providing further information on the experimental design and rationale. For example, ‘vvDD-IL-15-Ra’ is mentioned without a full explanation of the treatment in the results section, and while the authors have previously published on this virus, it should not be assumed knowledge and therefore it’s somewhat sudden mention in the result section may confuse readers. I acknowledge that this is briefly highlighted in the introduction, but further rationale is required, namely for the selection of IL-15.

I also have some concerns regarding the data presented and conclusions generated. The overarching aim of the study is to determine how the combination of OV and fereroptosis-inducer erastin can modulate anti-tumour immunity. The authors use assays to assess tumour cell death in vitro, and also perform in vivo studies to quantify tumour growth and immune cell phenotyping. However, this analysis falls short. For example, in Figure 6, RTPCR is used to assessed CD4 and CD8 gene expression but this does not quantify subsets of T cells, and the authors do not specify which populations are being analyzed via RTPCR. I am also not convinced that erastin has any synergistic role with VVDDIL15Ra, as these groups appear similar in most analysis shown. While the authors claim enhanced IFNg (Figure 5D), it’s unclear which cells are responsible for producing this cytokine and whether it has direct role in lowering tumour burdens. Although the authors quantify the number of IFNg+CD8+ T cells in Figure 7, further detail is required on the context in which IFNg was detected – was this in situ? After in vitro stimulation? What frequency of the CD8+ T cells were expressing IFNg across treatment groups? The study would benefit by filling in these ‘gaps’ and provide a more cohesive story.

Reproducibility

It does not appear that the experiments have been repeated more than once. Could the authors include the number of experiments that the data represents? In the case of tumour experiments, n=5-7 is a low number of mice to deduce any statistical significance, and indeed, all of the error bars overlap in the case of Figure 4B. Individual mice must be shown in all figures. This needs to be corrected in Figure 7.

Additional comments

The authors often write statements without references e.g lines 67-71. The authors often describe results without references to the Figures e.g lines 318-329. The text also reads quite colloquially and does not conform to strong scientific writing. For example, line 214 “as of today”, line 320 “The OV stimulated most types of “good” immune cells as expected” as so forth.  

Author Response

Authors’ Replies to the Reviewers

The reviewers’ original comments are in black font and authors’ responses are in blue font.

Reviewer #1:

The study by Liu et al aims to assess the combined effect of oncolytic virus and ferroptosis-induced death on anti-tumour immunity. Whilst the topic is interesting, the manuscript is not clearly defined and presents major flaws that must be addressed:

Data and rationale

Overall, the figures are easy to read, however the results could be improved by providing further information on the experimental design and rationale. For example, ‘vvDD-IL-15-Ra’ is mentioned without a full explanation of the treatment in the results section, and while the authors have previously published on this virus, it should not be assumed knowledge and therefore it’s somewhat sudden mention in the result section may confuse readers. I acknowledge that this is briefly highlighted in the introduction, but further rationale is required, namely for the selection of IL-15.

Answer: Thank you for pointing out this issue. We added one new paragraph and a few new sentences to the last paragraph of the Introduction to explain the rationale for why vvDD-IL-15Ra was chosen for this study.

“I also have some concerns regarding the data presented and conclusions generated. The overarching aim of the study is to determine how the combination of OV and ferroptosis-inducer erastin can modulate anti-tumour immunity. The authors use assays to assess tumour cell death in vitro, and also perform in vivo studies to quantify tumour growth and immune cell phenotyping. However, this analysis falls short. For example, in Figure 6, RT-PCR is used to assessed CD4 and CD8 gene expression but this does not quantify subsets of T cells, and the authors do not specify which populations are being analyzed via RT-PCR. I am also not convinced that erastin has any synergistic role with VVDD-IL15Ra, as these groups appear similar in most analysis shown.”

Answer: We fully agree with you that more a complete analysis could be performed. However, our analyses of tumor size, mouse survival (Fig. 4B and C; Fig. 5A and B), ELISpot assay (Fig. 5D), and immune cell phenotyping (Fig. 7) clearly indicated a statistical difference between mice treated with monotherapies (either erastin or OV) or the dual treatment (both OV and erastin). Therefore, OV and erastin worked together to improve therapeutic efficacy in two tumor models (either additively or synergistically). We had stated in the Conclusion of the original version that this was “synergistic” (only once). We changed this to “work additively or synergistically,” as this is not defined yet. Thank you!

Your suggestions are great ones. Our combination of an OV with ferroptosis is a novel strategy. This study is the first to use such a combination, and we clearly showed that this is a viable combinatorial strategy for improved therapeutic efficacy. Many future studies from us and other groups are needed to explore and elucidate mechanisms of action, interplay, or crosstalk between ferroptosis, OV, cancer cells, and the tumor microenvironment. We hope to understand much more in separate future studies. While the authors claim enhanced IFNg (Figure 5D), it’s unclear which cells are responsible for producing this cytokine and whether it has direct role in lowering tumour burdens. Although the authors quantify the number of IFNg+CD8+ T cells in Figure 7, further detail is required on the context in which IFNg was detected – was this in situ? After in vitro stimulation? What frequency of the CD8+ T cells were expressing IFNg across treatment groups? The study would benefit by filling in these ‘gaps’ and provide a more cohesive story.

Answer: Fig. 5D showed the results of IFN-g ELISpot assay using splenocytes consisting of both lymphocytes and NK cells. Both cell types may contribute to the antitumor activity in vivo. For Figure 7A, the detection of IFN-g+CD8+ T cells was in situ, as the tumor tissues were processed into single cells for flow cytometry directly (without in vitro manipulations). These results clearly demonstrated that only two groups of treatments, OV alone or OV + erastin dual therapy, generated significant quantities of IFN-g+CD8+ T cells. In addition, there were more IFN-g+CD8+ T cells in the dual therapy group (p ≤ 0.05). We also analyzed other types of immune parameters such as the ratio of CD8+ T cells/Treg, a good indicator of antitumor activity.

Reproducibility

It does not appear that the experiments have been repeated more than once.

(1). Could the authors include the number of experiments that the data represents?

Answer: Yes, reproducibility is one of the most important aspects of scientific research. To make sure that our results are reproducible, (1) we utilized two different tumor models: the MC38 colon cancer model (Fig. 5) and the Hepa1-6 liver tumor model (Fig. 4). (2). The animal experiments with the  MC38 colon tumor model (Fig. 5) have been performed with one pilot experiment and repeated twice with a complete set of groups (a total of three times). The other tumor model (Hepa1-6) has been performed only once. These facts have been added to the figure legends.

(2). In the case of tumour experiments, n=5-7 is a low number of mice to deduce any statistical significance, and indeed, all of the error bars overlap in the case of Figure 4B. Individual mice must be shown in all figures. This needs to be corrected in Figure 7.

You are correct that n = 5-7 is a relatively low number. However, we had consulted our biostatistician before the in vivo experiments were designed, and based on preliminary data, it was predicted that in animal experiments with n = 5, our expected results would be statistically significant. Eventually, our real experimental results confirmed that.

We replotted the data with individual curves in Fig. 4B. As we can see, the difference was relatively small within the same group, with a few exceptions. Among the erastin-treated mice, one showed a relatively smaller inhibition when compared to others in the same group. In the OV-treated mice, the inhibition effects in two mice were smaller than those in the remaining three mice. Because of these relatively small variations, the experiment was not only highly statistically significant, but significant differences were also observed in the dual therapy group when compared to any of the other groups.

We are not able to replot Figure 7 since IT management at the original host institution blocked our access to the data. The senior investigators and first author have all left the original institution (IT network run through the hospital system University of Pittsburgh Medical Center [UPMC]). We have the raw data stored on both the host computer system and a thumb drive. The original institution locked our thumb drive and we no longer have access to the original host computer system. The long approval procedure to restore access prevents us from re-organizing the plots for this revision. We sincerely apologize to you for this.

With that said, our original plots are in a different format and these data show a statistical difference between important groups.

Additional comments

The authors often write statements without references e.g lines 67-71. The authors often describe results without references to the Figures e.g lines 318-329.

We referenced the figures in those special contexts. Some text in the mentioned paragraphs has been rewritten to better clarify the statement. The total number of cited references increased from 43 to 50.

The text also reads quite colloquially and does not conform to strong scientific writing. For example, line 214 “as of today”, line 320 “The OV stimulated most types of “good” immune cells as expected” as so forth.

A native English speaker has now gone through the whole manuscript and polished it extensively to correct some of the expressions and improved its readability.

Reviewer 2 Report

The authors in this report investigated a combination therapy with an ferroptosis enhancer and OV on antitumor immunity. They found interesting results that combination therapy promoted antitumor immunity through advancing activated dendritic cells in the tumor tissue and by increasing CD8 T cells.

Effects of erastin or OV alone also shows limited effects in providing antitumor immunity but in combination it enhances the protective effects.

The study done is interesting and results are clearly defined but please address the following concerns:

  1. The mechanistic approach to link immune cells with the combination therapy is not clearly discussed.
  2. Page 1, line 39, add ‘as’ to such as.
  3. Line 100, correct the word combined to combination.
  4. Line 106, please add the reference.
  5. Please enlarge all the figures for clarity.
  6. Line 420, correct the word indirect.

Author Response

Reviewer #2:

The authors in this report investigated a combination therapy with an ferroptosis enhancer and OV on antitumor immunity. They found interesting results that combination therapy promoted antitumor immunity through advancing activated dendritic cells in the tumor tissue and by increasing CD8 T cells.

Effects of erastin or OV alone also shows limited effects in providing antitumor immunity but in combination it enhances the protective effects.

We thank you for your critical and positive evaluation of our study.

The study done is interesting and results are clearly defined but please address the following concerns:

  1. The mechanistic approach to link immune cells with the combination therapy is not clearly discussed.

We expanded our discussion on this in both the Introduction and Discussion.

  1. Page 1, line 39, add ‘as’ to such as.

Done. Thanks!

  1. Line 100, correct the word combined to combination.

Corrected. Thanks!

  1. Line 106, please add the reference.

We cited three references for the sources of these cell lines (new ref #37-39).

  1. Please enlarge all the figures for clarity.

In the revised version, we enlarged the figures. All figures are in TIF format and thus the publisher can easily adjust the sizes of the figures if needed.

  1. Line 420, correct the word indirect.

It has been corrected. Thank you again.

Round 2

Reviewer 1 Report

I thank the authors for responding to the original comments. However, I am concerned about the missing raw data in Figure 7.  The manuscript will be fit for publication only if this data is obtained and evidence of the data is presented to the journal as a means of upholding scientific standards and integrity.

Author Response

Thank you for your great suggestions. We have now converted the data presentation in Figure 7 to a format you have asked. In addition, we have improved the Introduction by reorganizing the paragraphs and polishing some sentences with help from a native English speaker.  Thank you again as your suggestions helped improve the quality of our manuscript.